# α-NiS/g-C₃N₄ Nanocomposites for Photocatalytic Hydrogen Evolution and Degradation of Tetracycline Hydrochloride

Huajin Qi [1,2,3,†], Chenyu Wang [2,†], Luping Shen [2], Hongmei Wang [2,*], Yuan Lian [3], Huanxia Zhang [3], Hongxia Ma [2], Yong Zhang [1] and Jin Zhong Zhang [4,*]

1   Key Laboratory of Advanced Textile Materials and Manufacturing Technology of the Ministry of Education, College of Textile Science and Engineering, Zhejiang Sci-Tech University, Hangzhou 310018, China; 202130202147@mails.zstu.edu.cn (H.Q.); zhangyong@zstu.edu.cn (Y.Z.)
2   Jiaxing Key Laboratory of Molecular Recognition and Sensing, College of Biological, Chemical Sciences and Engineering, Jiaxing University, Jiaxing 314001, China; 18991724674@163.com (C.W.); 13732590360@163.com (L.S.); jxmahx@mail.zjxu.edu.cn (H.M.)
3   College of Material and Textile Engineering, Jiaxing University, Jiaxing 314001, China; hnlianyuan@126.com (Y.L.); zhanghuanxia818@163.com (H.Z.)
4   Department of Chemistry and Biochemistry, University of California, Santa Cruz, CA 95064, USA
*   Correspondence: hongmei256@163.com (H.W.); zhang@ucsc.edu (J.Z.Z.)
†   These authors contributed equally to this work.

**Abstract:** α-NiS/g-C₃N₄ nanocomposites were synthesized and used for photocatalytic hydrogen (H₂) evolution and tetracycline hydrochloride (TC) degradation. The fabricated nanocomposites were characterized by XRD, XPS, SEM, TEM, UV-vis DRS, TRPL, and PEC measurements. Photocatalytic studies show that the hydrogen generation rate of the 15%-α-NiS/g-C₃N₄ nanocomposite reaches 4025 µmol·g⁻¹·h⁻¹ and TC degradation rate 64.6% within 120 min, both of which are higher than that of g-C₃N₄. The enhanced performance of the nanocomposite is attributed to the formation of a heterojunction between α-NiS and g-C₃N₄ that enhances visible light absorption, promotes the separation and transfer of charges, and inhibits the recombination of carriers. The photocatalytic mechanism of the α-NiS/g-C₃N₄ heterojunction nanocomposite is discussed in terms of relevant energy levels and charge transfer processes.

**Keywords:** photocatalysis; g-C₃N₄; surface modification; photodegradation; hydrogen evolution





## 1. Introduction

g-C₃N₄ is a promising material for photocatalytic applications in environmental purification, CO₂ reduction, hydrogen evolution, organic synthesis, and sterilization [1–7]. However, due to the rapid recombination of photogenerated electron and hole pairs, the photocatalytic efficiency of g-C₃N₄ is low, and modification is usually required to improve its performance [8]. Strategies for modification include morphology manipulation and metal or non-metal element doping [1,2,9–14]. However, the synthetic conditions for modification are often harsh, and the improvement of photocatalytic performance is limited [15].

One of the most effective strategies to improve photocatalytic hydrogen production is to load a cocatalyst on the surface of g-C₃N₄ to provide more active sites, rapidly separate photogenerated electrons and holes, and reduce the recombination probability [16]. The cocatalyst is low in content and does not affect the crystal shape and size of g-C₃N₄. Common cocatalysts include noble metals or non-noble metals. Supported noble metal cocatalysts, such as Pt, Au, Pd, and Ag, significantly improve the photocatalytic activity of g-C₃N₄, but their high price and scarcity greatly limit their application in practice [17].

As non-precious metal cocatalysts, metal sulfides, such as NiS [18,19], NiS₂ [20], MoS₂ [21–23], Ag₂S [24], and CoSₓ [25], have been widely reported for their application

in photocatalytic hydrogen production and pollutant degradation. For example, Yu et al. prepared amorphous $MoS_x$ (a-$MoS_x$) nanoparticles modified g-$C_3N_4$ photocatalyst [23]. Compared with pure g-$C_3N_4$, the photocatalytic hydrogen production performance of 3 wt%-$MoS_x$/g-$C_3N_4$ is improved by 91 times. This is attributed to the amorphous $MoS_x$ cocatalyst being conducive to the transfer of photogenerated electrons and effectively reducing the recombination of photogenerated carriers. Similarly, Jiang et al. designed $Ag_2S$/g-$C_3N_4$ photocatalyst and demonstrated a photocatalytic hydrogen production rate of 10 µmol·h$^{-1}$ for g-$C_3N_4$ loaded with 5 wt% $Ag_2S$, two orders of magnitude higher than that of pure g-$C_3N_4$ [24]. These studies show that metal sulfides are promising low-cost cocatalysts.

Among the metal sulfide cocatalysts, nickel sulfide (NiS) exhibits complicated structure, composition, and magnetic phase behavior [26–29]. For instance, NiS has two crystal structures: hexahedron NiS ($\alpha$-NiS) and rhombohedron NiS ($\beta$-NiS) [30]. The $\alpha$-NiS is beneficial to decompose $H_2O$ into $H^+$ and $OH^-$ [31], and $\beta$-NiS shows better conductivity than other phases of NiS [32]. In previous research, we have studied the influence of cocatalysts with different crystalline phases, $\alpha$-NiS-$\beta$-NiS and $\beta$-NiS, on hydrogen production using $Cd_{0.5}Zn_{0.5}S$ as a photocatalyst [33]. The 17%-$\alpha$-NiS-$\beta$-NiS/$Cd_{0.5}Zn_{0.5}S$ nanocomposite showed a photocatalytic $H_2$ evolution rate of 3113.0 µmol·h$^{-1}$·g$^{-1}$, better than that of 17%-$\beta$-NiS/$Cd_{0.5}Zn_{0.5}S$. These results showed that moderate loading of NiS cocatalyst can greatly enhance the photocatalytic hydrogen production activity of $Cd_{0.5}Zn_{0.5}S$, and the enhancement depends on the different NiS crystalline phases.

For g-$C_3N_4$, most research has focused on the development of new cocatalysts to improve their photocatalytic activity, and the influence of $\alpha$-NiS cocatalyst has not been systematically studied. In this work, $\alpha$-NiS/g-$C_3N_4$ nanocomposite photocatalysts were prepared for hydrogen generation with triethanolamine as a sacrificial agent and photodegradation of tetracycline hydrochloride (TC). Their crystal structure, morphology, light-harvesting capacity, and surface chemical states were characterized. The 15%-$\alpha$-NiS/g-$C_3N_4$ showed an optimal photocatalytic hydrogen rate of 4025 µmol·h$^{-1}$·g$^{-1}$, which was 35.7 times as high as that of pure g-$C_3N_4$, and the photocatalytic degradation rate of TC reached 64.6% under visible light irradiation, respectively. The recyclability of as-synthesized 15%-$\alpha$-NiS/g-$C_3N_4$ nanocomposite was also evaluated. Based on ESR experiments, a possible photocatalytic mechanism is proposed.

## 2. Results and Discussion

### 2.1. Synthesis, Structure, and Morphology

The synthesis route of $\alpha$-NiS/g-$C_3N_4$ photocatalyst is shown schematically in Figure 1a. Firstly, dicyandiamide and ammonium chloride were selected as the precursors to obtain g-$C_3N_4$ in thin layers. With g-$C_3N_4$ as the base material, $\alpha$-NiS was grown on g-$C_3N_4$ with close contact. The crystal structure of the obtained sample was determined by XRD. As shown in Figure 1b, the diffraction peaks at 30.3°, 34.8°, 46.0°, and 53.7° correspond to the (100), (101), (102), and (110) crystal planes of NiS, respectively, which are consistent with the standard card (PDF No. 02-1280), indicating that the NiS have been successfully prepared. Meanwhile, the diffraction peak at ~27.3° corresponds to the (002) crystal plane, which is caused by the 2D g-$C_3N_4$ interlayer stacking reflection [34]. However, the diffraction peak (~13.1°) of the (100) crystal plane corresponding to 2D g-$C_3N_4$ is disappeared due to the 2D ultra-thin structure of 2D g-$C_3N_4$ [35,36]. The typical peaks of $\alpha$-NiS and g-$C_3N_4$ are observed for the $\alpha$-NiS/g-$C_3N_4$ composites with no impurity phases. In addition, there is no shift for the diffraction peak position of $\alpha$-NiS or g-$C_3N_4$ in the $\alpha$-NiS/g-$C_3N_4$ composites, suggesting the crystal structure remains during the process of synthesis. In addition, the spacing between layers of 2D g-$C_3N_4$ is about 0.335 nm, and the average crystallite size of NiS is calculated to be 17.5 nm according to Scherrer's equation [37].

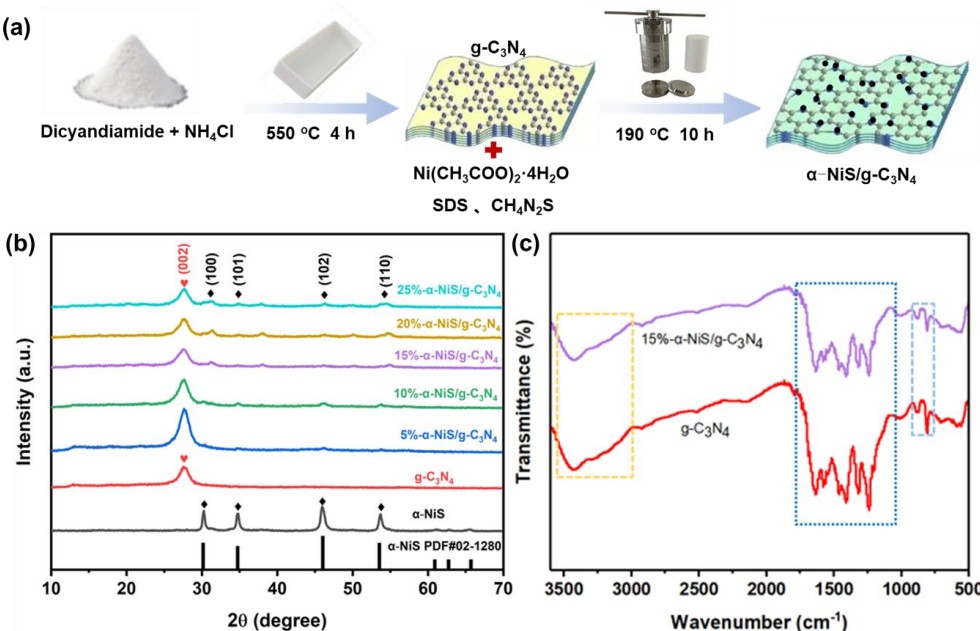

**Figure 1.** (**a**) Synthesis route for the preparation of α-NiS/g-C$_3$N$_4$ photocatalysts; (**b**) XRD patterns of α-NiS, g-C$_3$N$_4$, and α-NiS/g-C$_3$N$_4$ composites; (**c**) FT-IR spectra of g-C$_3$N$_4$, and 15%-α-NiS/g-C$_3$N$_4$.

The pristine g-C$_3$N$_4$ and 15%-α-NiS/g-C$_3$N$_4$ were characterized by FT-IR. As observed in Figure 1c, the characteristic peaks at 810 and 878 cm$^{-1}$ are attributed to the triazine units and N-H, respectively. The absorption bands around 1110–1730 cm$^{-1}$ are related to the C-N heterocycle stretching vibration modes, and the broad peaks between 3000–3500 cm$^{-1}$ are N-H and O-H stretching vibrations [38]. In particular, the FT-IR spectrum of 15%-α-NiS/g-C$_3$N$_4$ nanocomposite was the same as that of 2D g-C$_3$N$_4$, indicating that g-C$_3$N$_4$ in 15%-α-NiS/g-C$_3$N$_4$ retains its chemical structure, consistent with XRD analysis.

The morphologies of g-C$_3$N$_4$, α-NiS, and 15%-α-NiS/g-C$_3$N$_4$ were characterized by SEM. As shown in Figure 2a,b, g-C$_3$N$_4$ showed a folded sheet structure, and NiS showed nanoparticles (NPs). In Figure 2c, both g-C$_3$N$_4$ and α-NiS are observed in the 15%-α-NiS/g-C$_3$N$_4$ nanocomposite, and α-NiS and g-C$_3$N$_4$ seem to be in close contact.

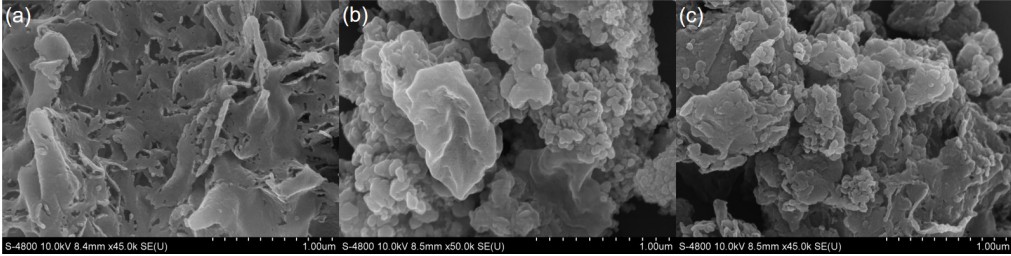

**Figure 2.** SEM images of (**a**) g-C$_3$N$_4$, (**b**) α-NiS, and (**c**) 15%-α-NiS/g-C$_3$N$_4$.

The morphologies of all photocatalysts were further determined by HRTEM. As shown in Figure 3a, many α-NiS NPs (dark color) are decorated on the surface of transparent g-C$_3$N$_4$ nanosheets. Though no lattice fringe of g-C$_3$N$_4$ was observed, its wavy structure is discernable. The d-spacing of 0.258 nm corresponds to the (101) lattice plane of α-NiS (PDF No. 02-1280) [39]. These results suggest that α-NiS NPs are in good contact with g-C$_3$N$_4$ forming heterojunctions. In addition, the element composition of 15%-α-NiS/g-C$_3$N$_4$ was analyzed with HAADF-STEM EDX mapping images, as shown in Figure 3b. The elements C, N, Ni, and S are shown and evenly distributed in 15%-α-NiS/g-C$_3$N$_4$, which further confirms the successful preparation of α-NiS/g-C$_3$N$_4$ nanocomposite.

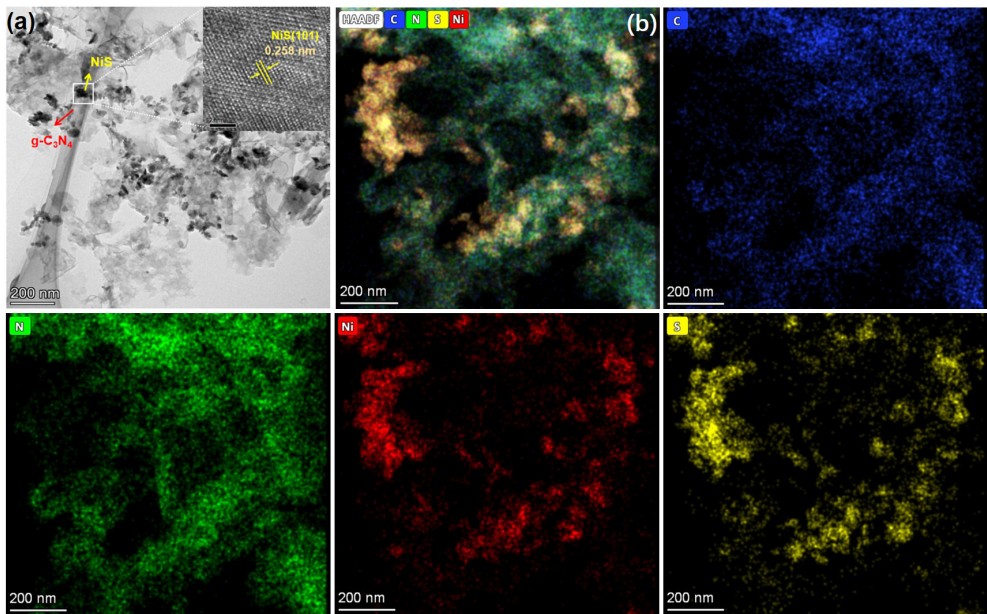

**Figure 3.** (**a**) HRTEM image of 15%-$\alpha$-NiS/g-C$_3$N$_4$ and (**b**) HAADF-STEM EDX elemental mapping images.

XPS analysis of $\alpha$-NiS/g-C$_3$N$_4$ was conducted with results shown in Figure 4. In Figure 4a, the C 1s spectra of $\alpha$-NiS/g-C$_3$N$_4$ are deconvoluted into three peaks centered at 287.9, 286.5, and 284.8 eV, corresponding to sp$^2$ hybridized carbon (N=C-N), $\pi$–$\pi$* of adventitious carbon cycles and graphitic carbon (C–C), respectively [40]. As for N 1s shown in Figure 4b, its high-resolution XPS spectra can be fitted into three peaks centered at 398.5, 400.3, and 404.3 eV, which arose from sp$^2$ hybridized N (C–N–C, N$_2$C), tertiary N of three-fold coordination (N–C$_3$, N$_3$C) and the charge effect or the localization of the positive charge in the heterocyclic ring, respectively [41]. The connection of C–N–C and N–C$_3$ constitutes a tri-s-triazine conjugated skeleton of g-C$_3$N$_4$. Figure 4c shows the high-resolution of Ni 2p fitted with two spin-orbit doublets and two shakeup satellite peaks. The dominant peaks of 855.4 and 873.1 eV are related to Ni$^{2+}$, and the lower peaks of 852.6 and 870.0 eV originated from Ni$^{\delta+}$ with very small positive charges of NiS [42–44]. Two main shake-up satellite peaks at 860.7 and 879.3 eV were also observed. From Figure 4d, the S 2p spectra show two peaks at 162.3 and 161.0 eV, related to S 2p$_{3/2}$ and S 2p$_{1/2}$, respectively [45].

### 2.2. Optical and Photoelectrochemical Properties

The light absorption of a photocatalyst directly affects its photocatalytic performance. The UV-vis DRS spectra of the prepared photocatalysts are shown in Figure 5a. Compared with g-C$_3$N$_4$, the absorption edges of the $\alpha$-NiS/g-C$_3$N$_4$ nanocomposite were redshifted with increased $\alpha$-NiS content. For $\alpha$-NiS/g-C$_3$N$_4$, light absorption is stronger than that of g-C$_3$N$_4$ in the 500–800 nm range, which is due to the introduction of NiS that contributes to visible light absorption. Based on the absorption spectra, E$_g$ was calculated to be 2.76 and 2.52 eV for g-C$_3$N$_4$ and 15%-NiS/g-C$_3$N$_4$ using Equation (1), respectively [46,47].

$$\text{Band gap} = \frac{1240}{\text{Wavelength}} \tag{1}$$

The Mott–Schottky (M−S) plot was measured at 1000 Hz in 0.10 M Na$_2$SO$_4$ solution to determine the conduction band (CB) of the g-C$_3$N$_4$. In Figure 5b, the positive slope of the Mott–Schottky plot indicates that g-C$_3$N$_4$ is an n-type semiconductor. The flat band potential (V$_{fb}$) of g-C$_3$N$_4$ was calculated to be −0.37 V vs. NHE at pH 7 based on the linear potential curves. The bottom of the CB is −0.1 V compared to V$_{fb}$ for n-type

semiconductors, and the $E_{CB}$ of g-C$_3$N$_4$ is $-0.47$ V vs. NHE at pH 7 [48]. The valence band (VB) edge potential was determined to be 2.29 V vs. NHE at pH 7 based on $E_{CB} = E_{VB} - E_g$.

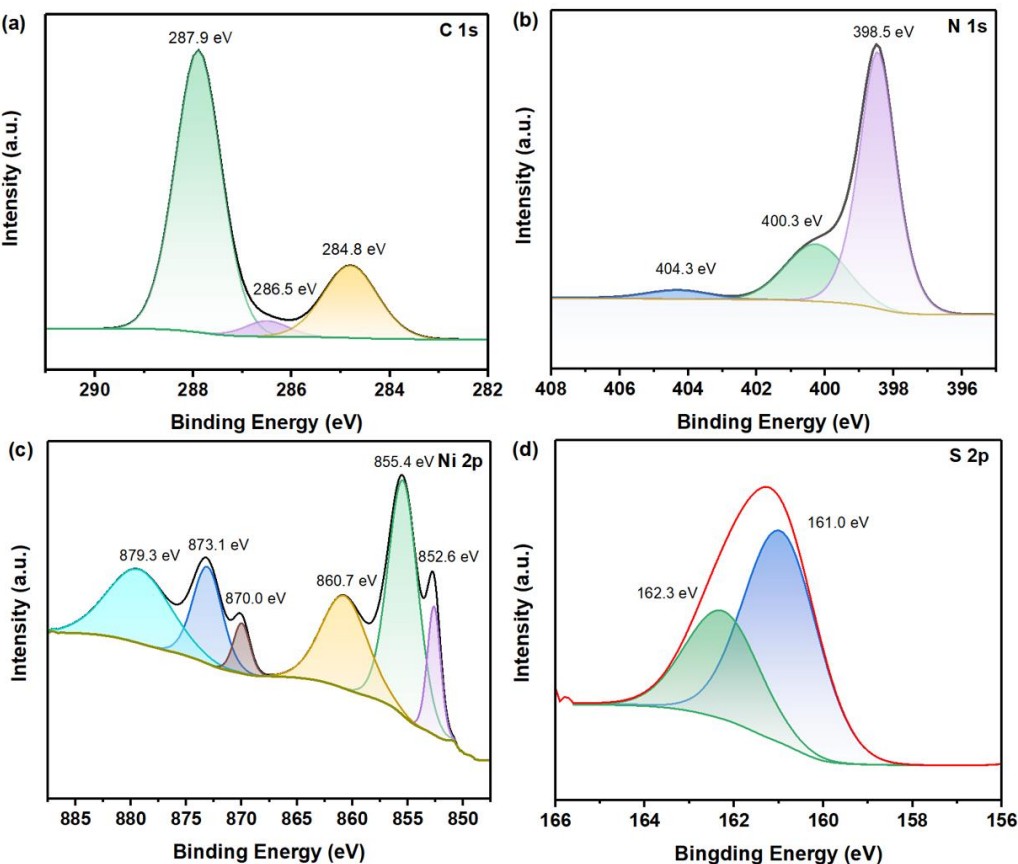

**Figure 4.** The XPS spectra of (**a**) C 1s, (**b**) N 1s, (**c**) Ni 2p, and (**d**) S 2p in α-NiS/g-C$_3$N$_4$.

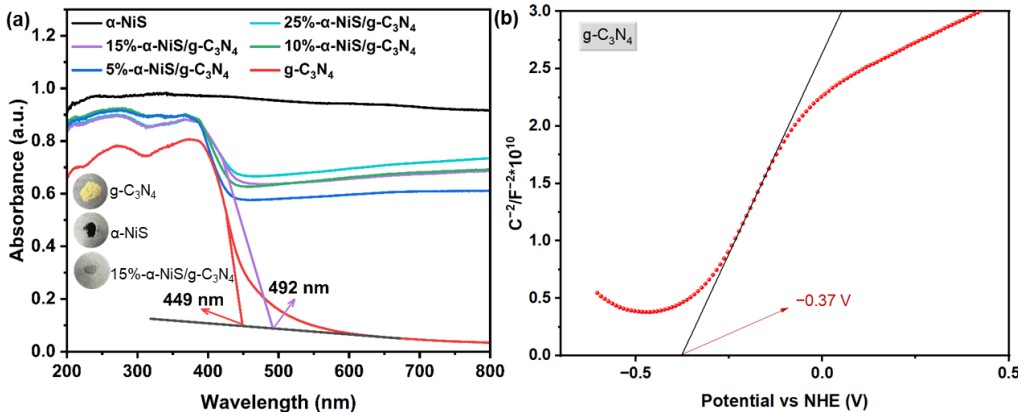

**Figure 5.** (**a**) UV-vis DRS of α-NiS, g-C$_3$N$_4$, and α-NiS/g-C$_3$N$_4$ nanocomposite; (**b**) Mott–Schottky plot of g-C$_3$N$_4$.

Electrochemical studies of g-C$_3$N$_4$, 5%-α-NiS/g-C$_3$N$_4$, 10%-α-NiS/g-C$_3$N$_4$, and 15%-α-NiS/g-C$_3$N$_4$ were conducted. Figure 6a shows the transient photocurrent–time (I–t) curves obtained. The results show that 15%-α-NiS/g-C$_3$N$_4$ has the highest photocurrent intensity, indicating that α-NiS loading leads to more efficient electron transfer [49]. Therefore, 15%-α-NiS/g-C$_3$N$_4$ has the optimal hydrogen evolution performance. Electrochemical impedance spectroscopy (EIS) provides more evidence for the charge transfer efficiency of catalysts. In Figure 6b, the 15%-α-NiS/g-C$_3$N$_4$ nanocomposite has the minimum resistance curve, indicating minimum resistance [50].

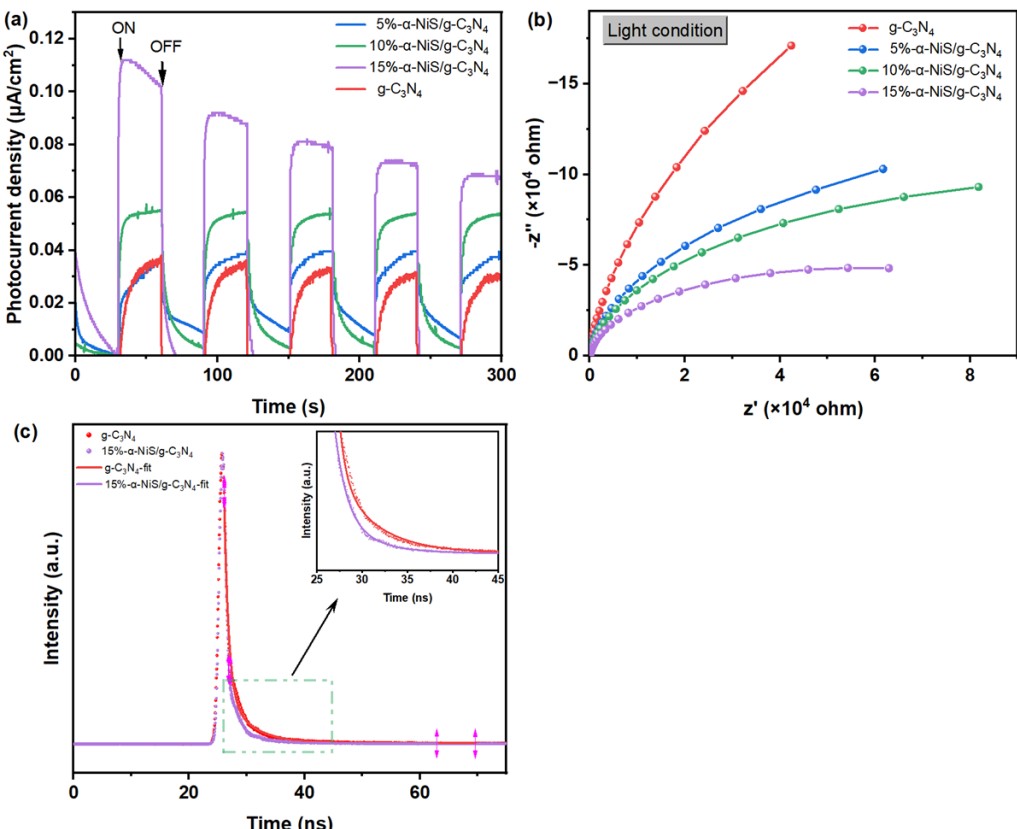

**Figure 6.** (**a**) Transient photocurrent and (**b**) Nyquist impedance plots of the different as-prepared samples; (**c**) TRPL profiles of g-C$_3$N$_4$ and 15%-α-NiS/g-C$_3$N$_4$.

Figure 6c shows the TRPL profiles of α-NiS, g-C$_3$N$_4$, and 15%-α-NiS/g-C$_3$N$_4$, which are used to determine the average lifetime of photogenerated carriers τ using the following Equation (2) [34,51,52]:

$$\tau = \frac{A_1\tau_1^2 + A_2\tau_2^2}{A_1\tau_1 + A_2\tau_2} \tag{2}$$

where A$_1$ and A$_2$ are the constants obtained after fitting the decay curves with τ$_1$ and τ$_2$ representing fast and slow components. The average lifetime of the charge carriers of 15%-α-NiS/g-C$_3$N$_4$ (τ = 1.35 ns) is longer than that of pure g-C$_3$N$_4$ (τ = 0.76 ns). Therefore, α-NiS loaded on the surface of g-C$_3$N$_4$ lengthened the average carrier lifetime, which is desired for photocatalytic applications [53]. The result is consistent with that of electrochemical studies.

### 2.3. Evaluation of Photocatalytic Performance

2.3.1. Photocatalytic H$_2$ Generation and Stability of α-NiS/g-C$_3$N$_4$ Nanocomposites

Photocatalytic H$_2$ evolution over g-C$_3$N$_4$, α-NiS, and different proportions of α-NiS/g-C$_3$N$_4$ nanocomposites was evaluated under visible light irradiation. Figure 7a shows the photocatalytic H$_2$ evolution from pure g-C$_3$N$_4$. With the increase of α-NiS in the α-NiS/g-C$_3$N$_4$ nanocomposite, the performance of H$_2$ evolution is improved. When the mass ratio of α-NiS and g-C$_3$N$_4$ is 15 wt%, the H$_2$ production amount reaches 20,125 μmol·g$^{-1}$, which is about 35.7 times higher than that of pure g-C$_3$N$_4$. With further increase of α-NiS loading in α-NiS/g-C$_3$N$_4$, the photocatalytic performance is decreased because excessive α-NiS would suppress the H$_2$ production activity [20,54]. Hence, α-NiS with moderate loading can accelerate electron separation, which will provide more reaction sites and promote H$_2$ production. Figure 7b shows the hydrogen evolution rates of different photocatalysts. The photocatalytic hydrogen evolution rates of pure g-C$_3$N$_4$, 5%-α-NiS/g-C$_3$N$_4$, 10%-α-NiS/g-C$_3$N$_4$, 15%-α-NiS/g-C$_3$N$_4$, 20%-α-NiS/g-C$_3$N$_4$, and 25%-α-NiS/g-C$_3$N$_4$ are 113 μmol·g$^{-1}$·h$^{-1}$, 2959 μmol·g$^{-1}$·h$^{-1}$, 3165 μmol·g$^{-1}$·h$^{-1}$, 4025 μmol·g$^{-1}$·h$^{-1}$,

3598 μmol·g$^{-1}$·h$^{-1}$, 3047 μmol·g$^{-1}$·h$^{-1}$. Too high a loading of α-NiS likely decreased the light absorption of g-C$_3$N$_4$, which will reduce the photocatalytic H$_2$ evolution activity. In addition, the cycling stability of the 15%-α-NiS/g-C$_3$N$_4$ nanocomposite was measured every 5 h as one cycle, and the results are shown in Figure 7c. The photocatalytic H$_2$ production activity of 15%-α-NiS/g-C$_3$N$_4$ nanocomposite does not significantly decrease, indicating good photocatalytic stability. The photocatalyst was characterized using XRD after the stability test. As shown in Figure 7d, no evident change in the XRD pattern of the 15%-α-NiS/g-C$_3$N$_4$ nanocomposite was observed after the photocatalytic reaction, indicating no structural change.

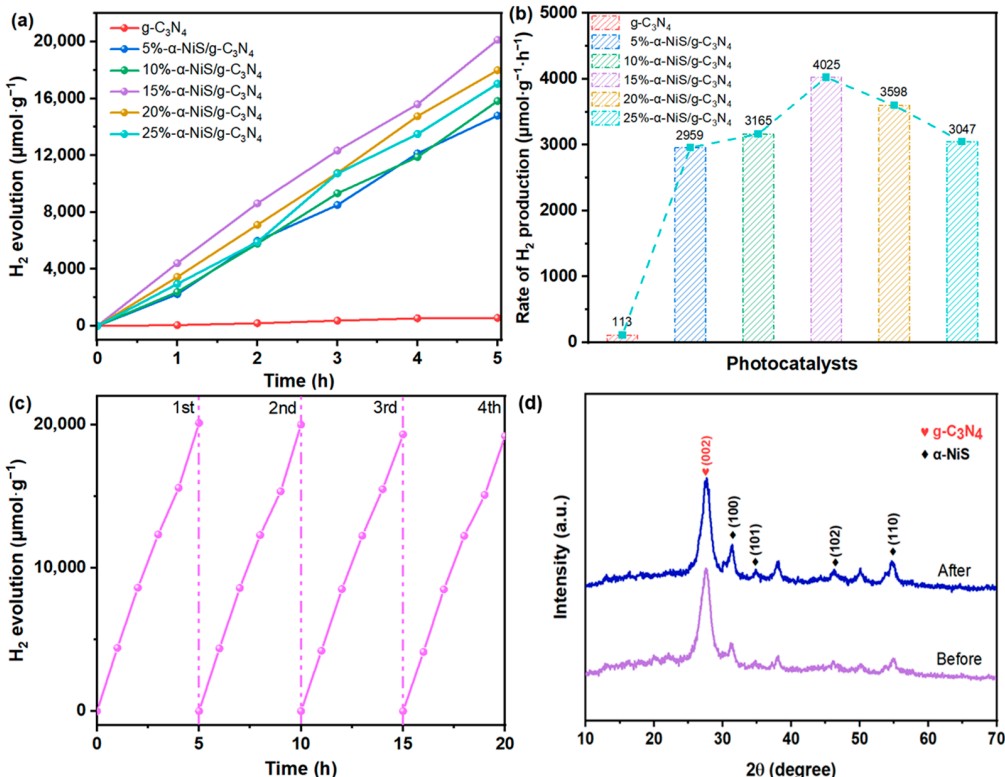

**Figure 7.** The photocatalytic hydrogen production activities of photocatalysts (**a**,**b**) and recycling test of 15%-α-NiS/g-C$_3$N$_4$ (**c**) under visible light irradiation (λ ≥ 400 nm) in the presence of TEOA; (**d**) XRD pattern of the 15%-α-NiS/g-C$_3$N$_4$ nanocomposite before and after the photocatalytic reaction.

### 2.3.2. Photocatalytic TC Degradation

Figure 8a shows the TC removal rate curve in terms of C/C$_0$ as a function of light irradiation time. After adsorption and photocatalytic reaction, the removal rate of TC for the optimized 15%-α-NiS/g-C$_3$N$_4$ was 67.6%. The degradation ratio of TC through the photocatalytic process for 15%-α-NiS/g-C$_3$N$_4$ was 64.6%. The result suggests that adsorption played a negligible role in TC removal. The improved photocatalytic activity contributed to the loading of α-NiS cocatalyst that effectively promoted the separation of charges. Figure 8b shows the absorption spectra of TC with time evolution over 15%-α-NiS/g-C$_3$N$_4$. The photocatalytic performance of 15%-α-NiS/g-C$_3$N$_4$ was monitored by measuring the absorbance peak at 375 nm, which decreased with increasing the irradiation time [53].

As shown in Figure 8c, the kinetics of photodegradation is described by the pseudo-first-order equation ($-\ln(C/C_0) = kt$), suggesting the photocatalytic degradation activity of the photocatalyst [55]. In the inset of Figure 8c, the apparent kinetic constants (k) of TC degradation for pure g-C$_3$N$_4$ and 15%-α-NiS/g-C$_3$N$_4$ were 0.00199 min$^{-1}$ and 0.00968 min$^{-1}$, respectively. The k value of 15%-α-NiS/g-C$_3$N$_4$ was about 4.86 times as high

as that of the pure g-C$_3$N$_4$, indicating that the 15%-$\alpha$-NiS/g-C$_3$N$_4$ nanocomposite possessed enhanced photocatalytic performance. Figure 8d shows the reusability of the prepared 15%-$\alpha$-NiS/g-C$_3$N$_4$ nanocomposite for photocatalytic TC degradation. After four operations, the photocatalytic activity was not decreased obviously, and the photocatalytic activity of 15%-$\alpha$-NiS/g-C$_3$N$_4$ still remained at about 64.5% for TC, and the removal rate was decreased only by 2.2% compared with the initial value. The results show that the as-prepared 15%-$\alpha$-NiS/g-C$_3$N$_4$ has good reusability and stability in the photocatalytic process.

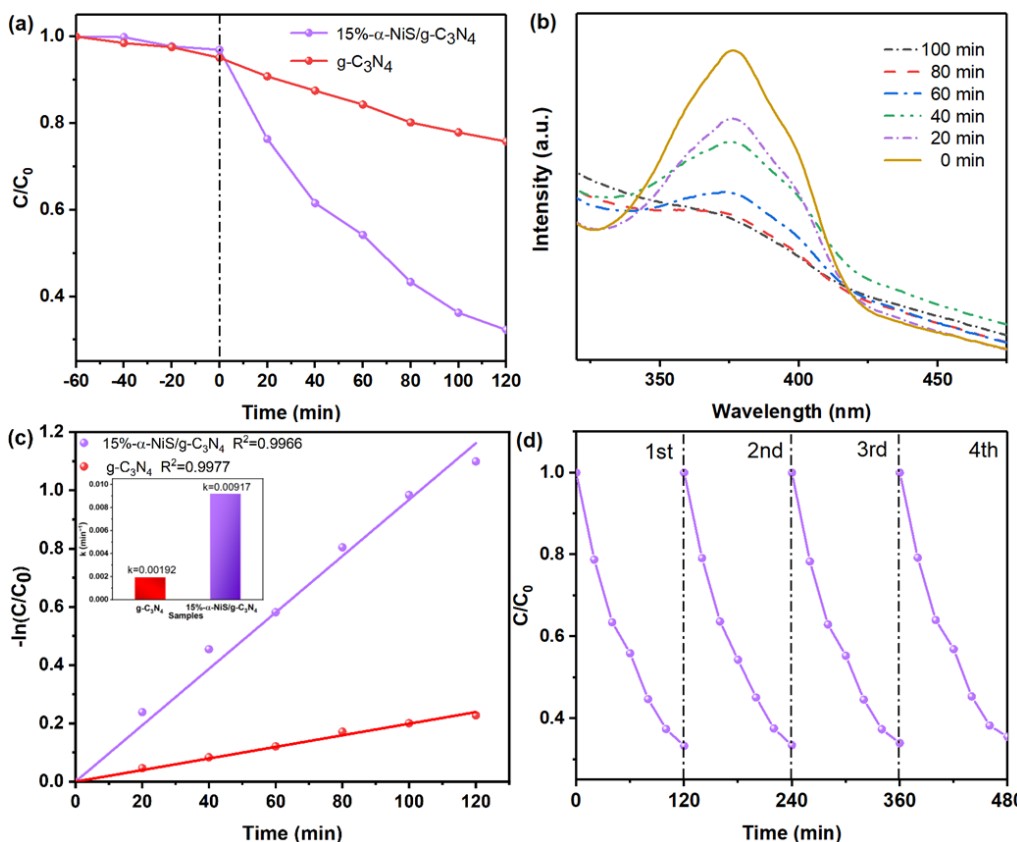

**Figure 8.** (**a**) Photocatalytic degradation of different $\alpha$-NiS/g-C$_3$N$_4$ photocatalysts for TC (20 mg L$^{-1}$), (**b**) absorption spectra for TC degradation over 15%-$\alpha$-NiS/g-C$_3$N$_4$, (**c**) the corresponding $-\ln(C/C_0)$ vs. irradiation time plots, and (**d**) reusability of 15%-$\alpha$-NiS/g-C$_3$N$_4$.

The results of the radical scavenging study are shown in Figure 9a. The activities of hydroxyl radicals ($\cdot$OH), photogenerated holes (h$^+$), and superoxide anion radical ($\cdot$O$_2^-$) were inhibited by isopropyl alcohol (IPA), ethylene diamine tetra acetic acid (EDTA), and 1,4-benzoquinone (BQ), respectively [56]. The photocatalytic degradation activities are significantly reduced after the addition of IPA and BQ, which shows that $\cdot$OH and $\cdot$O$_2^-$ play a dominant role in the photocatalytic TC degradation. The inhibition effect of EDTA on the degradation efficiency for TC was weaker than that of BQ and IPA, suggesting that the contribution of h$^+$ was less than $\cdot$OH and $\cdot$O$_2^-$. In addition, ESR was used to further investigate the active components involved [57]. In Figure 9b,c, no obvious signals of DMPO-$\cdot$OH and DMPO-$\cdot$O$_2^-$ adducts can be probed under dark conditions, but signals for these two adducts appeared under light irradiation with intensity increasing with irradiation time. The ESR results show that both $\cdot$OH and $\cdot$O$_2^-$ radicals were generated in the photocatalytic process with the 15%-$\alpha$-NiS/g-C$_3$N$_4$, which is in agreement with the results of quenching experiments [58,59].

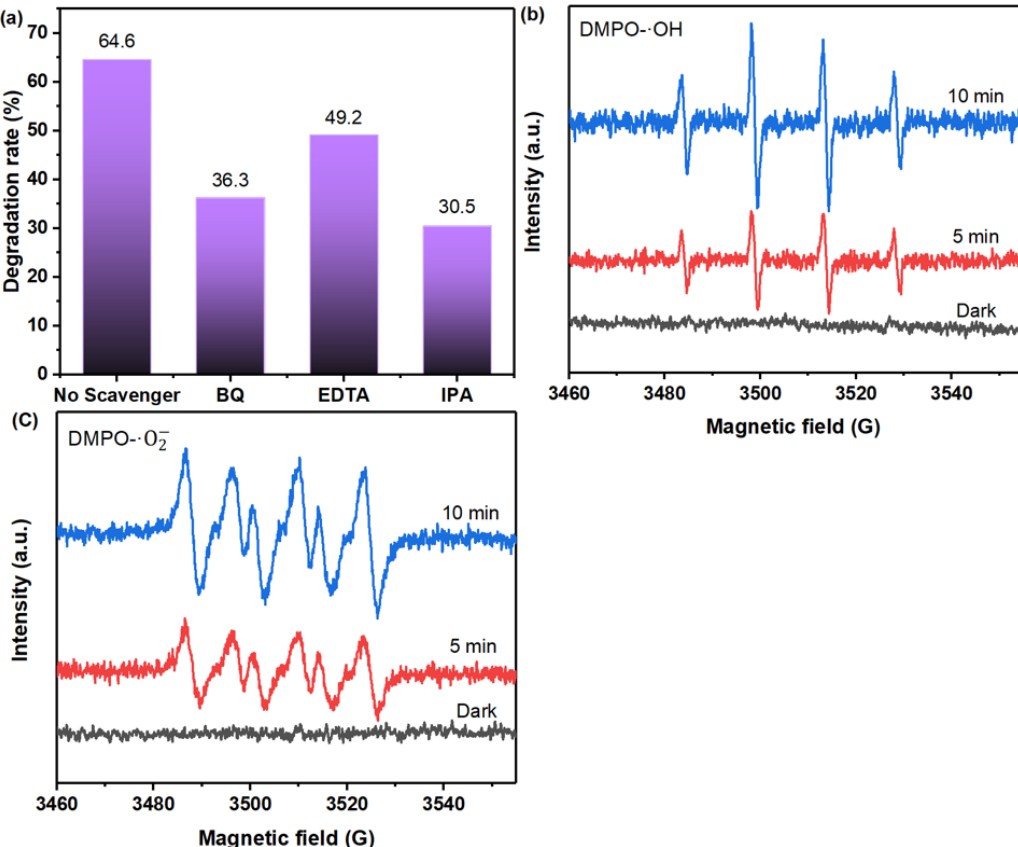

**Figure 9.** (**a**) The effects of different scavengers on photocatalysis; ESR spectra of (**b**) DMPO-OH and (**c**) DMPO-$O_2^-$ over the 15%-α-NiS/g-$C_3N_4$ nanocomposite.

### 2.4. Proposed Photocatalytic Mechanism

Based on UV-Vis DRS and Mott–Schottky results, the conduction band (CB) of g-$C_3N_4$ is −0.47 V (vs. NHE), and the valence band (VB) of g-$C_3N_4$ is 2.29 V (vs. NHE). As illustrated in Figure 10a, light excitation of g-$C_3N_4$ results in photogenerated electrons ($e^-$) in the CB and holes ($h^+$) in the VB. The photogenerated $e^-$ in the CB of g-$C_3N_4$ is expected to transfer to α-NiS. $O_2$ in the solution can react with electrons in the CB of g-$C_3N_4$ or be transferred to α-NiS to form an active species ·$O_2^-$, which can react with TC causing its degradation. Meanwhile, the photogenerated holes in the VB of g-$C_3N_4$ react with $H_2O$ to form ·OH, which can also react with TC to result in its degradation.

Figure 10b shows an illustration of the photocatalytic hydrogen evolution with EY for sensitization. The bandgap of 2.76 eV of g-$C_3N_4$ allows charge carrier generation. The $e^-$ in the CB of g-$C_3N_4$ is transferred to α-NiS, and $h^+$ can accumulate in the VB of g-$C_3N_4$, allowing efficient carrier separation. In addition, the EY molecules are excited to form $EY^{1*}$, which can lead to the formation of a triplet excited state ($EY^{3*}$) via an intersystem crossing. In the TEOA solution, $EY^{3*}$ can be reduced, forming free radical $EY^-$ with strong reducibility. The $EY^-$ can transfer electrons to the CB of g-$C_3N_4$, and the regenerated EY takes part in the next electron transfer cycle. The photogenerated $e^-$ in both g-$C_3N_4$ and EY can be transferred to α-NiS to reduce $H^+$ to produce $H_2$. Finally, the $h^+$ accumulated in the VB of g-$C_3N_4$ is consumed by TEOA.

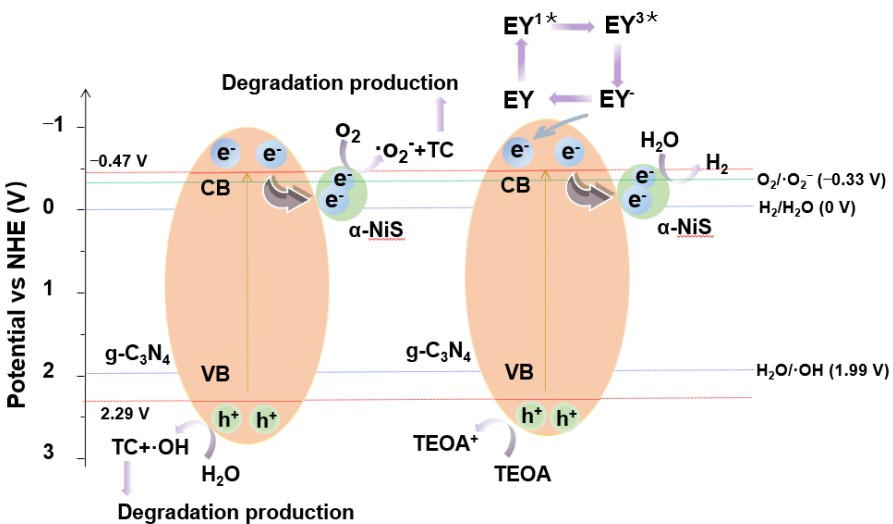

**(a) Photocatalytic Degradation   (b) Photocatalytic Hydrogen**

**Figure 10.** Schematic diagram of the charge transfer pathways of the $\alpha$-NiS/g-C$_3$N$_4$ nanocomposite under visible light irradiation for photocatalytic degradation of TC (**a**) and photocatalytic hydrogen evolution (**b**).

## 3. Materials and Methods

### 3.1. Chemicals

Dicyandiamide (C$_2$H$_4$N$_4$), ammonium chloride (NH$_4$Cl), nickel acetate tetrahydrate (Ni(CH$_3$COO)$_2$·4H$_2$O), and thioacetamide (CH$_3$CSNH$_2$), sodium dodecyl sulfate (CH$_3$-(CH$_2$)$_{11}$OSO$_3$Na), triethanolamine (C$_6$H$_{15}$NO$_3$) were supplied by Aladdin Reagent Company (Shanghai, China). Ethyl alcohol (C$_2$H$_5$OH) was purchased from Sinopharm Chemical Reagent Co., Ltd. (Shanghai, China) and deionized water was used throughout this work.

### 3.2. Photocatalyst Preparation

3.2.1. Synthesis of g-C$_3$N$_4$

A total of 1.00 g dicyandiamide was ground with 10.0 g ammonium chloride, and the mixture was then transferred to the corundum crucible and covered and calcined in a Muffle furnace. The Muffle furnace was heated to 550 °C at 3 °C/min and maintained for 4 h, and g-C$_3$N$_4$ nanosheets were obtained when the furnace was cooled to room temperature.

3.2.2. Synthesis of $\alpha$-NiS/g-C$_3$N$_4$ Photocatalysts

The as-prepared 0.100 g g-C$_3$N$_4$ was added into a beaker filled with 60 mL of ethanol for ultrasonic treatment for 40 min. Then, an appropriate amount of Ni(CH$_3$COO)$_2$·4H$_2$O, sodium dodecyl sulfate (SDS), and CH$_4$N$_2$S were added into the solution with continuous stirring for one hour. Then, the mixture solution was transferred to a 100 mL Teflon-lined autoclave and heated to 190 °C in an oven for 10 h. Finally, the product was cooled to room temperature and washed with deionized water and ethanol three times before being dried in a vacuum oven at 60 °C overnight. x%-$\alpha$-NiS/g-C$_3$N$_4$ nanocomposites were prepared by varying the amount of NiS (x = 5, 10, 15, 20, 25), where x represented the mass ratio of $\alpha$-NiS to g-C$_3$N$_4$.

### 3.3. Characterizations

The crystal structure of the obtained samples was characterized by an X-ray diffractometer (XRD-7000, Shimadzu Corporation, Kyoto, Japan). FT-IR spectra of the samples were obtained using Fourier transform infrared spectrometer (Nicolet iS50, Thermo Fisher Scientific, Waltham, MA, USA). The morphology and microstructure were determined by emission electron microscopy (S-4800, Hitachi, Tokyo, Japan) and a high-resolution transmission electron microscope (Talos F200X, Thermo Fisher Scientific, Waltham, MA,

USA). The surface composition and chemical state of the samples were measured by an X-ray photoelectron spectrometer (ESCALAB 250Xi, Thermo Fisher Scientific, Waltham, MA, USA). Ultraviolet-visible diffuse reflection spectra (UV-vis DRS) were obtained using a UV-vis near-infrared spectrometer (Cary 5000, Agilent, Santa Clara, CA, USA). The time-resolved PL (TRPL) profiles were determined with a time-resolved PL spectrometer (FLS980, Edinburgh Instrument, Livingston, UK).

### 3.4. Evaluation of Photoelectrochemical Performance

Transient photocurrent and electrochemical impedance spectroscopy (EIS) measurements were conducted using a standard three-electrode system. First, 5.00 mg of the photocatalyst sample was added into 1.00 mL of mixture solution containing 800 μL isopropyl alcohol and 200 μL deionized water. Then, 40–60 μL of 5 vol% Nafion solution was added into the above-mentioned mixture solution. The mixture was dispersed evenly by ultrasonication and then spun onto FTO glass. The FTO glass coated with as-prepared samples was used as the working electrode, the standard Ag/AgCl electrode as the reference electrode, and the platinum electrode was used to construct a three-electrode system. The electrolyte was 0.100 M $Na_2SO_4$ solution, and the photoelectrochemical (PEC) performance of the sample was evaluated by an electrochemical workstation (CHI-660E, Shanghai Chenhua, Shanghai, China).

### 3.5. Evaluation of Photocatalytic Performance

The photocatalytic $H_2$ production and TC degradation were used to evaluate the photocatalytic performance of the prepared samples. In the experiment of $H_2$ production, 10 mg photocatalyst was ultrasonically dispersed into 70 mL aqueous solution including triethanolamine (TEOA, 15 vol%) with 25 mg of EY as a sacrificial agent, then the reactor was sealed, pumped, and purged with nitrogen for 30 min, and the oxygen in the reactor was emptied. A 300 W Xe lamp ($\lambda > 400$ nm) was placed at the top of the photoreactor, and the light source was turned on to start the photocatalytic reaction. Throughout the reaction, cooling water is passed through the reactor jacket to maintain a constant reaction temperature (15 °C). The reaction lasted for five hours, and samples were taken once every hour. The generated $H_2$ was measured by gas chromatography (5 Å molecular sieve-packed column). The detector was a thermal conductivity detector (TCD), and the carrier gas was high-purity nitrogen.

In addition, TC was selected as the target pollutant to evaluate the photodegradation performance of the α-NiS/g-$C_3N_4$ nanocomposites. Typically, a 30 mg photocatalyst was ultrasonically dispersed in 50 mL TC aqueous solution (20 mg/L) and stirred in the dark for 60 min to ensure that TC reached adsorption–desorption equilibrium. Then, the 300 W Xe lap source ($\lambda > 400$ nm) is activated to start the photodegradation reaction, and cooling water is injected to maintain a constant reaction temperature throughout the process. In the process of photodegradation, the photocatalyst was removed by 0.22 μm polyether sulfone membrane after taking samples at regular intervals. The absorbance of TC in the filtrate was measured at the maximum absorption wavelength $\lambda_{max} = 375$ nm by a UV-visible spectrophotometer.

## 4. Conclusions

α-NiS/g-$C_3N_4$ nanocomposites were constructed and evaluated for photocatalytic applications. By comparing x%-α-NiS/g-$C_3N_4$ (x = 5, 10, 15, 20, and 25) with various loading amounts of α-NiS, the photocatalytic $H_2$ evolution, and TC degradation first increased with the optimal α-NiS content for α-15%-α-NiS/g-$C_3N_4$ and then decreased with further increment of α-NiS content. The 15%-α-NiS/g-$C_3N_4$ showed the highest photocatalytic activities with an $H_2$ evolution rate of 4025 $\mu mol \cdot g^{-1} \cdot h^{-1}$ and 67.6% for TC removal rate. The heterojunctions formed between α-NiS and g-$C_3N_4$ increased absorption of visible light, facilitated charge carrier separation, and prolonged the lifetime of charge carriers, thus improving the photocatalytic efficiency. The ESR spectra and active species trapping

experiments indicated that h$^+$, ·OH, and ·O$_2$$^-$ all contributed to TC photodegradation. A possible photocatalytic mechanism was proposed for the α-NiS/g-C$_3$N$_4$ nanocomposites based on their relative electronic energy levels and associated charge transfer processes.

**Author Contributions:** H.Q. and C.W.: writing—original draft, writing—review and editing; L.S.: methodology, data curation; H.W., H.M. and J.Z.Z.: conceptualization, methodology, supervision, writing—review and editing; Y.L., H.Z. and Y.Z.: conceptualization, funding acquisition, methodology, supervision. All authors have read and agreed to the published version of the manuscript.

**Funding:** This research work was financially supported by the Basic Public Research Project of Zhejiang Province (No. LGC20B050014) and the Scientific Training Program for College Students of Jiaxing University (No. 8517221490).

**Data Availability Statement:** Data that support the findings of this study are included within the article.

**Conflicts of Interest:** The authors declare no conflict of interest.

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
