# Peer review of "α-NiS/g-C3N4 Nanocomposites for Photocatalytic Hydrogen Evolution and Degradation of Tetracycline Hydrochloride"

_catalysts, doi:10.3390/catal13060983_

Round 1

Reviewer 1 Report

This manuscript reported the α-NiS/g-C3N4 nanocomposites for photocatalytic hydrogen (H2) evolution and tetracycline hydrochloride (TC) degradation under light irradiation. The optimal 15%-α-NiS/g-C3N4 sample exhibit higher activity than the bulk C3N4 due to the enhanced absorption capacity of visible light, promoted the separation and transfer of charges, and inhibited the recombination of carriers. The catalysts are well characterized, and conclusions are supported by various experimental data. I would like to accept this manuscript after minor revision.

1. Fig. 5a, the absorption edge of 15%-α-NiS/g-C3N4 is not 575nm, and this is wrong. Pleas correct it.

2. Fig.5d is suggested to move to Fig.6 since the TRPL is employed to investigate the charge carriers’ separation.

3. The catalyst after the stability test should be characterized to confirm the unchanged structure.

4. Related references should be cited: 10.1002/anie.202204563; 10.1016/j.cej.2023.142022; 10.1016/j.apcatb.2020.119789

Some typos should be corrected.

Reviewer 2 Report

The development of a stable and highly active photocatalyst for wastewater treatment is an important research area currently. In this study α-NiS/g-C3N4 nanocomposites have been developed as photocatalysts for production of hydrogen gas and photodegradation of organic pollutants. The developed photocatalysts have been characterized by enough number of advanced techniques. However, the discussion on characterization of the photocatalysts is weak.

1.       Abstract: “Photocatalysis studies show……….” It should be “Photocatalytic studies…..”

2.       Similarly, in Introduction, “…………photocatalysis applications in environmental………” should be “………photocatalytic application…………”

3.       In introduction, line 33, “……..photogenerated electron and hole pairs, its efficiency of……”. It should be ““……..photogenerated electron and hole pairs, the efficiency of……”.

4.       Introduction, line 36, “However, the synthesis conditions……….” It should be “synthetic conditions…”

5.       Introduction, line 48, “For example, Yu et al. prepared amorphous molybdenum sulfide modified g-C3N4 photocatalyst [22]” Reference 22 does not report this study. It is irrelevant.

6.       Introduction, line 49, “Compared with pure g-C3N4, the photocatalytic hydrogen………..” There is no reference for this report.

7.       Introduction, line 59, “……high temperature hexahedron (α-NiS) and low temperature rhombohedron (β-NiS)……” What is meant by high temperature and low temperature structures??? It is an awkward sentence.

8.       Line 69, “In our previous research, the effects of…………..” This sentence is awkward. It must be rephrased accordingly. Similarly, the next sentence “More recently, a variety………….” must also be corrected/rephrased.

9.       Line 74, “……..and the removal rate for TC is apt to achieve……….”. It must be corrected.

10.   Line 75, “And the cycling stability……….” It is an awkward sentence.

11.   The discussion on XRD is not complete. The crystallite size of particles using Sherrer equation should be calculated. Similarly, the crystal structure/unit cell with lattice parameters should also be calculated. It can be done by non-linear method of analysis as reported in Phys. Scr. 96 (2021) 125707 https://doi.org/10.1088/1402-4896/ac237a

12.   There is no discussion on elemental mapping (Fig 3b).

13.   In UV-DRS, the band gap has been calculated for g-C3N4 only. Band gap for composite can also be calculated. For example, See UV-DRS analysis reported in Surfaces and Interfaces 30 (2022) 101846 https://doi.org/10.1016/j.surfin.2022.101846 and New J. Chem., 2022, 46, 2224–2231

14.   There is no reference for discussion on Electrochemical studies (Fig 6).

15.   Fig 8c: The intercept of the straight line should be set as zero while applying the trend line.

16.   “With the increase of α-NiS loading in α-NiS/g-C3N4, the photocatalytic performance is decreased, because excessive α-NiS would suppress the H2 production activity”. The reason given is not appropriate. Reason for the decrease in catalytic performance with an increase in NiS loading with suitable references must be given.

English must be improved. 

Author Response

please see attached response file.

Round 2

Reviewer 2 Report

The manuscript has been revised accordingly.